# URANIA: Differentially Private Insights into AI Use

**Daogao Liu**
liudaogao@gmail.com

**Edith Cohen**
edith@cohenwang.com

**Badih Ghazi**
badihghazi@gmail.com

**Peter Kairouz**
kairouz@google.com

**Pritish Kamath**
pritishk@google.com

**Alexander Knop**
aaknop@gmail.com

**Ravi Kumar**
ravi.k53@gmail.com

**Pasin Manurangsi**
pasin@google.com

**Adam Sealfon**
adamsealfon@google.com

**Da Yu**
dayuwork@google.com

**Chiyuan Zhang**
chiyuan@google.com

Google Research

## Abstract

We introduce URANIA, a novel framework for generating insights about LLM chatbot interactions with rigorous differential privacy (DP) guarantees. The framework employs a private clustering mechanism and innovative keyword extraction methods, including frequency-based, TF-IDF-based, and LLM-guided approaches. By leveraging DP tools such as clustering, partition selection, and histogram-based summarization, URANIA provides end-to-end privacy protection. Our evaluation assesses lexical and semantic content preservation, pair similarity, and LLM-based metrics, benchmarking against a non-private method inspired by CLIO (Tamkin et al., 2024). Moreover, we develop a simple empirical privacy evaluation that demonstrates the enhanced robustness of our DP pipeline. The results show the framework's ability to extract meaningful conversational insights while maintaining stringent user privacy, effectively balancing data utility with privacy preservation.

## 1 Introduction

Large language models (LLMs) have become ubiquitous tools used by hundreds of millions of users daily through chatbots such as ChatGPT, Gemini, Claude.ai, and DeepSeek. It is valuable for both LLM platform providers and the general public to understand the high-level use cases for which these chatbots are employed. For example, such information could help platform providers detect if the chatbots are used for purposes that violate safety policies. However, while LLM platform providers have access to data of user queries, there are serious privacy concerns about revealing information about user queries.

Recently, Tamkin et al. (2024) proposed CLIO, a system that uses LLMs to aggregate insights while preserving user privacy, building on Zheng et al. (2023); Zhao et al. (2024) (see §E for more details). CLIO provides insights about how LLM chatbots (Claude.ai in their case) are used in the real world and visualizes these patterns in a graphical interface. Subsequently, Handa et al. (2025) applied CLIO to more than four million Claude.ai conversations, to provide insight into which economic tasks are performed by or with chatbot help.

CLIO is based on heuristic privacy protections. It starts by asking an LLM to produce individual summaries and to strip them out of private information from original user queries. The queries are clustered based on their individual summaries, and only large clusters are finally released, along with a joint summary of all the queries within each cluster. Furthermore, these clusters are organized into a hierarchy.

In other words, the privacy protection that CLIO provides is loosely based on the notion of $k$-anonymity (Sweeney, 2002). However, the exact privacy guarantee of CLIO is hard to formalize, as it depends on properties of the LLM used in the method. For example, it relies on prompts such as "*When answering, do not include any personally identifiable information (PII), like names, locations, phone numbers, email addresses, and so on. When answering, do not include any proper nouns*". While such approaches can work reasonably well in practice, they rely on

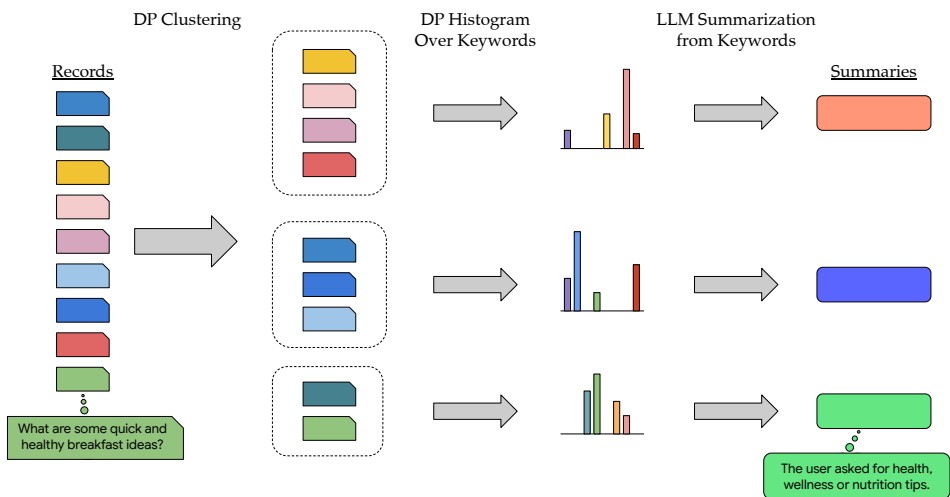

Figure 1: Illustration of our proposed URANIA method.

heuristic properties of the LLMs, and hence may or may not work well with future versions of the LLMs, thereby making such a system hard to maintain. To counter this, CLIO also includes a *privacy auditor*, also based on an LLM via a prompt that starts as "*You are tasked with assessing whether some content is privacy-preserving on a scale of 1 to 5. Here's what the scale means: ...*", while acknowledging that no such system can be perfect. In fact, Tamkin et al. (2024) explicitly note that "*since Clio produces rich textual descriptions, it is difficult to apply formal guarantees such as differential privacy and k-anonymity*". Thus, a natural question arises:

> *Is it possible to obtain acceptable utility for the task of summarizing user queries,*
> *with formal end-to-end DP guarantees?*

Differential privacy (DP) (Dwork et al., 2006) is a powerful mathematical notion that is considered the gold standard for privacy protection in data analytics and machine learning, and had been widely applied in practice (see, e.g., Desfontaines, 2021).

**Our Contributions.** We propose URANIA, a novel framework for summarizing user queries to an LLM chatbot with a formal end-to-end DP guarantee, that *does not rely* on any heuristic properties of the underlying LLM used (§4); our approach uses LLMs for keyword extraction, and utilizes existing tools including DP clustering, partition selection and histogram release (see §2 for descriptions of these tools). En route,

In §2: we introduce the formal problem statement of *text summarization* of user queries and discuss the DP tools used in the work,

In §3: we propose a simplified version of CLIO that we refer to as SIMPLE-CLIO (similar to CLIO, this does not satisfy any formal privacy guarantees). We consider this simplified version since the implementation of CLIO is not publicly available, and also, CLIO has additional aspects where it obtains many "facets" for each query, e.g., a "concerning content" score, language of query, etc., which we ignore for simplicity.

In §5: we propose a family of evaluation metrics to compare the quality of any two query summarization methods, and we use these to compare the performance of SIMPLE-CLIO and URANIA, and finally,

In §F: we propose a simple *empirical privacy evaluation* method using a membership inference-style attack to conceptually demonstrate that indeed URANIA is more robust than SIMPLE-CLIO.

Although we focus here on the application of summarizing user queries to an LLM chatbot, our techniques could be more generally applicable to summarizing arbitrary text corpora.[1]

---

[1]This might require a change to the specific prompts that we use.

We view the formalization of the text summarization problem and the design and evaluation of the first DP mechanism for it as our main contributions. We believe there are several avenues for improving such a system, either by better prompt engineering or by using better DP subroutines (e.g., for private clustering). Thus, we view our work as the first step in this promising and important research direction.

## 2  Formal Problem and Preliminaries

We formalize the "text summarization" problem by describing the (i) *input* to the problem, (ii) desired *output*, and (iii) approaches that we use for measuring *utility*.

**Input.** The input is a dataset $x = \{x_1, \dots, x_n\}$ of conversations, where each element $x_i \in \mathcal{X}$ represents a conversation between a user and an LLM.

**Output.** The desired output for our problem is $s = \{s_1, \dots, s_k\}$, consisting of *summaries* $s_j \in \mathcal{S}$. The set $\mathcal{S}$ is potentially infinite (e.g., the set of all strings in our case). The DP guarantee applies only to the release of this output set $s$ of summaries.

**Utility.** Evaluating the utility of privacy-preserving summaries presents unique challenges due to the unstructured nature of text data. For utility measurement purposes only, we produce a mapping $\varphi : x \to s$ so that $\varphi(x_i)$ is the summary with which the record $x_i$ is associated. Importantly, releasing this mapping would violate DP and hence is never returned—it serves solely as an internal mechanism for evaluation.

Rather than defining an explicit utility function, we adopt a comparative evaluation framework that assesses utility across multiple dimensions: (i) Lexical and semantic content preservation, (ii) Topic coverage and representation, (iii) Embedding-based semantic similarity, (iv) LLM-based assessment. This multifaceted approach allows us to quantify the privacy-utility trade-off across different privacy parameter settings and implementation configurations. We defer the detailed discussion of these evaluations and their results to §5.

### 2.1  Differential Privacy

All the datasets in this paper are collections of records. A randomized function that maps input datasets to an output space is referred to as a *mechanism*. For mechanism $\mathcal{M}$ and dataset $x$, we use $\mathcal{M}(x)$ to denote the random variable returned over the output space of $\mathcal{M}$. Two datasets $x$ and $x'$ are said to be *adjacent*, denoted $x \sim x'$, if, one can be obtained by adding or removing one record from the other; this is referred to as the "add-remove" adjacency. We consider the following notion of $(\varepsilon, \delta)$-*differential privacy (DP)*.

**Definition 2.1.** Let $\varepsilon > 0$ and $\delta \in [0, 1]$. A mechanism $\mathcal{M}$ satisfies $(\varepsilon, \delta)$-*differential privacy* ($(\varepsilon, \delta)$-DP for short) if for all adjacent datasets $x \sim x'$, and for any (measurable) event $E$ it holds that $\Pr[\mathcal{M}(x) \in E] \leq e^{\varepsilon} \Pr[\mathcal{M}(x') \in E] + \delta$.

We also say that a mechanism satisfies $\varepsilon$-DP if it satisfies $(\varepsilon, 0)$-DP. To prove privacy guarantees, we will use basic properties of DP stated in Proposition A.1 (in §A).

**DP Clustering.** Clustering is a central primitive in unsupervised machine learning (see, e.g., Ikotun et al., 2023). Abstractly speaking, a (Euclidean) clustering algorithm takes as input a dataset of vectors in $\mathbb{R}^d$, and returns a set $C \subseteq \mathbb{R}^d$ of $k$ "cluster centers" (for a specified parameter $k$), that are representative of the dataset; formalized for example, via the $k$-means objective (Lloyd, 1982). Clustering algorithms have been well-studied in the literature for several decades, and more recently with DP guarantees as well. We discuss more related work in §E. While our framework is compatible with any DP clustering algorithm, specifically, we use the implementation of DP clustering in the open-source Google DP library[2], as described in Chang & Kamath (2021); we denote the $(\varepsilon, \delta)$-DP instantiation of any such algorithm producing (up to) $k$ clusters as $\mathtt{DP-K-Means}_{k,\varepsilon,\delta}$.

---

[2]https://github.com/google/differential-privacy/tree/main/learning/clustering

**DP Histogram Release.** In the private histogram release problem over a set $\mathcal{B}$ of bins, the input is a multi-set $\{h_1, h_2, \ldots\}$, where each $h_i \subseteq \mathcal{B}$ with $|h_i| \leq k$. The desired output is a histogram that must be as close to $\sum_i h_i$ as possible, where we abuse notation to interpret $h_i$ as an indicator vector in $\{0,1\}^{\mathcal{B}}$.

There are multiple algorithms for this problem (e.g., the Laplace mechanism, Gaussian mechanism, etc.); in this paper, we use the discrete Laplace mechanism introduced by Ghosh et al. (2012), where $\mathsf{PHR}_{k,\varepsilon}$ is the instantiation of this algorithm that is $\varepsilon$-DP and tolerates removal of up to $k$ elements. See §A for formal details.

**Theorem 2.2** (Ghosh et al. (2012)). *For all integers $k > 0$ and $\varepsilon > 0$: $\mathsf{PHR}_{k,\varepsilon}$ satisfies $\varepsilon$-DP.*

**DP Partition Selection.** In the private partition selection problem over a (potentially infinite) set $\mathcal{B}$, the input is a multi-set $\{h_1, h_2, \ldots\}$ where each $h_i \subseteq \mathcal{B}$ with $|h_i| \leq k$. The desired output is a subset $S \subseteq \mathcal{B}$ such that $S \subseteq \bigcup_i h_i$ and $S$ is as large as possible. We use the partition selection algorithm introduced by Desfontaines et al. (2022); $\mathsf{PPS}_{k,\varepsilon,\delta}$ is the instantiation of this algorithm that is $(\varepsilon, \delta)$-DP. See §A for formal details.

**Theorem 2.3** (Desfontaines et al. (2022)). *For all integers $k > 0$, $\varepsilon > 0$ and $\delta \in [0,1]$: $\mathsf{PPS}_{k,\varepsilon,\delta}$ satisfies $(\varepsilon, \delta)$-DP.*

# 3 SIMPLE-CLIO

We present a non-DP (public) framework for hierarchical text summarization, which is a simplified version of CLIO (Tamkin et al., 2024). Our approach transforms a dataset $x = (x_1, \ldots, x_n) \in \mathcal{X}^n$ of conversations into high-level summaries $s = (s_1, \ldots, s_m) \in \mathcal{S}^m$ through three stages. The pseudocode can be found in Algorithm 4 (in §B) with the implementation details as discussed below.

1. **[Embedding Generation]** To use a clustering algorithm, we convert conversations to numerical vectors. A typical way to do this is to use pre-trained embedding models; specifically, we use `all-mpnet-base-v2` (Reimers & Gurevych, 2019; 2022). However, following Tamkin et al. (2024), we first map the conversation to a summary text before applying the embedding model; this ensures that conversations that are similar in content but structured differently get mapped to embeddings that are closer. We use an LLM to map each conversation to the summary text, using a prompt as described in §G.1.

2. **[Clustering]** We apply the standard $k$-means clustering (Lloyd, 1982) to group similar conversations based on their embeddings. We select $k$ to achieve an average cluster size of approximately 150 conversations. We assume $\mathrm{KMEANS}_k(\cdot)$ returns the centers of clusters and the assignments (points in each cluster).

3. **[Summary Generation]** For each cluster, we generate a representative summary by sampling both random conversations from the cluster and contrastive conversations near the cluster center. We then use an LLM to produce a summary that captures the main theme of the cluster. The prompt used for this step is provided in §G.2.

This framework provides an approach for organizing and summarizing large collections of conversations, enabling the identification of key topics without relying on manual annotation or predefined categories. However, as mentioned earlier, it does not satisfy any formal privacy guarantee. We next proceed to describing URANIA, our DP framework.

# 4 URANIA: A DP Framework for Text Summarization

We present URANIA, a framework for hierarchical text summarization (with Hierarchy and Simple Visualization presented in §H) that builds on the public framework while providing formal DP guarantees. Unlike SIMPLE-CLIO that is non-private, URANIA incorporates privacy-preserving mechanisms at critical steps to ensure that the resulting summaries do not leak sensitive information about individual conversations.

Our approach takes a dataset $x = (x_1, \ldots, x_n) \in \mathcal{X}^n$ of conversations, a predefined set $\mathcal{K}$ of keywords, a number $k \in \mathbb{N}$ of clusters, a number of keywords used for summary, and

---

**Algorithm 1** URANIA$_{k,t,\varepsilon_c,\varepsilon_{\text{hist}},\varepsilon_{\text{size}},\delta_c}$: Differentially Private Text Corpora Summarization

---

**Parameters:** Number of clusters $k$, number $t$ of keywords used for summaries, and cluster size threshold $\tau$.
**Parameters:** Privacy parameters $\varepsilon_c, \varepsilon_{\text{hist}}, \varepsilon_{\text{size}} > 0$, $\delta_c \in [0,1]$.
**Input:** Input dataset $x = (x_1, \ldots, x_n) \in \mathcal{X}^n$, keyword set $\mathcal{K}$
**Output:** High Level Summaries $s = (s_1, \ldots, s_k) \in \mathcal{S}^k$

// Step 1: Extract Embeddings from Conversations
$e \leftarrow \{\text{ExtractEmbeddings}(x_i) : i \in [n]\}$
// Step 2 (a): Cluster embeddings and assign records to cluster centers
$c \leftarrow \text{DP-KMEANS}_{k,\varepsilon_c,\delta_c}(e)$
Let $C_1, \ldots, C_k \subseteq \{x_1, \ldots, x_n\}$ such that $x_i \in C_j$ iff $j = \arg\min_{j'} \|c_{j'} - e_i\|_2$
// Step 3: Extract keywords
Let $K_j \leftarrow \varnothing$ for $j \in [k]$ ;                              // Initialize empty keyword sets
Let $C^{(\text{size})} \leftarrow \text{PHR}_{1,\varepsilon_{\text{size}}}(\{|C_i|\}_{i \in [k]})$ ;                    // Estimate sizes of clusters.
**for** $j \leftarrow 1$ **to** $k$ *such that* $C_j^{(\text{size})} \geq \tau$ **do**
 **if** $|C_j| = 0$ **then**
  $K_j \leftarrow \text{RandomKeywords}(\mathcal{K}, t)$ ;                    // Random keywords for empty clusters
 **else**
  // Step 3 (b): Keyword generation for qualifying clusters
  $\textbf{samples}_j \leftarrow \text{RandomSample}(C_j, m)$ ;        // Sample up to $m$ conversations from cluster
  Let $\textbf{relevant\_keywords}_j$ be an empty collection
  **for** $x \in \textbf{samples}_j$ **do**
   Append $\text{LLMSelectKeywords}(x, \mathcal{K}, 5)$ to $\textbf{relevant\_keywords}_j$;
         // Select up to 5 relevant keywords
  Let $r_{j,k}$ be the number of times a keyword $k \in \mathcal{K}$ is present in $\textbf{relevant\_keywords}_j$
  $\textbf{private\_hist}_j \leftarrow \text{PHR}_{5,\varepsilon_{\text{hist}}}(\{r_{j,k}\}_{k \in \mathcal{K}})$ ;                // Apply DP histogram mechanism
  $K_j \leftarrow \text{TopKeywords}(\textbf{private\_hist}_j, t)$ ;                            // Select top $t$ keywords
// Step 4: Generate summaries
$s \leftarrow \{\text{LLMSummarize}(K_j) : j \in [k]\}$
**return** $s$

---

privacy parameters $\varepsilon_c, \varepsilon_{\text{hist}}, \varepsilon_{\text{size}} > 0$ and $\delta_c \in [0,1]$. The choice of the set $\mathcal{K}$ of keywords is critical to the quality of the generated summaries; we discuss multiple strategies for this in §4.1. Our method produces high-level topical summaries $s = (s_1, \ldots, s_k) \in \mathcal{S}^k$ through the following privacy-preserving pipeline:

1. **[Embedding Generation]** This step is the same as in SIMPLE-CLIO.
2. **[Clustering]** Non-private clustering algorithms usually output both cluster centers and cluster assignments. However, cluster assignments are non-private for add-remove adjacency; hence, DP clustering algorithms output only cluster centers.
   (a) First, we generate cluster centers using an $(\varepsilon_c, \delta_c)$-DP k-means algorithm. Although we use the algorithm described in Chang & Kamath (2021), any DP clustering algorithm can be used here and the same privacy guarantees would follow.
   (b) Each conversation is assigned to its nearest cluster center.
3. **[Keyword Extraction]** For each cluster, we:
   - Apply DP to the cluster size with privacy parameter $\varepsilon_{\text{size}}$. If the noised size is smaller than a threshold $\tau$, we skip the cluster to avoid generating summaries that will not be meaningful.
   - Sample several conversations from the cluster. For empty clusters that pass the threshold due to noise, we randomly select keywords from the predefined set.
   - Use an LLM to extract relevant keywords from the predefined set $\mathcal{K}$ for each sampled conversation. For the purpose of DP, we restrict the number of keywords chosen per conversation to at most 5. See prompts in §G.3.
   - Apply a DP histogram mechanism with privacy parameter $\varepsilon_{\text{hist}}$ to identify the most frequent keywords.

- Test various settings of $\varepsilon_{\text{size}}$ and $\tau$ to evaluate the trade-off between privacy protection and utility.

4. **[Summary Generation]** For each cluster, we generate a high-level summary using only the selected keywords. See prompts in §G.4. We also discuss creating a hierarchy of summaries in §H.

**Theorem 4.1** (Privacy Guarantee). *Let $k, t \in \mathbb{N}$, $\varepsilon_c, \varepsilon_{\text{hist}}, \varepsilon_{\text{size}} > 0$, and $\delta_c \in [0, 1]$. Then* URANIA$_{k,t,\varepsilon_c,\varepsilon_{\text{hist}},\varepsilon_{size},\delta_c}$ *satisfies* $(\varepsilon_c + \varepsilon_{\text{hist}} + \varepsilon_{\text{size}}, \delta_c)$*-DP.*

*Proof.* The algorithm can be split into four parts: cluster center generation from the original dataset (steps 1-2(a)), estimating cluster sizes, an algorithm generating key words for a given list of cluster centers passing size test and the original dataset (step 3 (b)), and summary generation from a given set of keywords (step 4). We use DP properties (Proposition A.1).

The first part satisfies $(\varepsilon_c, \delta_c)$-DP since it consists of operations applied per record and subsequent application of the $(\varepsilon_c, \delta_c)$-DP instantiation of the DP-KMEANS algorithm. The second part is $\varepsilon_{\text{size}}$-DP provided that the set of cluster centers is fixed since it again applies per-record operations and an $\varepsilon_{\text{size}}$-DP histogram release algorithm. The third part is $\varepsilon_{\text{hist}}$-DP provided that the set of cluster centers is fixed since it again applies per-record operations and an $\varepsilon_{\text{hist}}$-DP histogram release algorithm, by parallel composition. Altogether, this means that the algorithm producing $K_1, \ldots, K_k$ is $(\varepsilon_c + \varepsilon_{\text{hist}} + \varepsilon_{\text{size}}, \delta_c)$-DP due to basic composition.

Finally, note that the third part is processing the output of the first two parts, so due to the post-processing property of DP, the entire algorithm satisfies $(\varepsilon_c + \varepsilon_{\text{hist}} + \varepsilon_{\text{size}}, \delta_c)$-DP. $\square$

**Remark 4.2.** The process of choosing the set of keywords $\mathcal{K}$ could itself consume some privacy budget, which needs to be accounted for separately. We discuss these methods next.

### 4.1 Choosing the Set of Keywords $\mathcal{K}$

The effectiveness of URANIA crucially depends on the quality of the keyword set $\mathcal{K}$ used during cluster summarization. We explore four approaches for constructing keyword sets $\mathcal{K}$, with varying privacy guarantees: namely, KwSet-TFIDF and KwSet-LLM satisfy DP, KwSet-Public uses a public dataset to obtain keywords, and KwSet-Hybrid uses a hybrid approach. We describe these methods informally below and formally in §C. We discuss some related work in §E.

**TF-IDF-Based Selection with Partition Selection (KwSet-TFIDF).** We compute the document frequency (DF) of all terms with additive Laplace noise to satisfy DP. Using these noisy DF values with token frequency (TF), we select 3–5 keywords for each conversation. We then apply partition selection (Theorem 2.3) to privately identify significant keywords, followed by LLM refinement to obtain ~200 final keywords (see §G.5 for prompt used for refining). This approach is better suited when direct LLM access to conversations raises additional privacy concerns.

**LLM-Based Selection with Partition Selection (KwSet-LLM).** We first use an LLM to select 3–5 keywords for each conversation independently. This produces a large candidate set of keywords across all conversations. We then apply partition selection to privately identify the most representative keywords while satisfying $(\varepsilon, \delta)$-DP. This yields several thousand keywords, which we refine using the LLM to produce a final set of ~200 keywords. See detailed prompts in §G.5.

**Iterative NLP-based Refinement (KwSet-Public).** As a non-private baseline, we first use multiple NLP techniques (Named Entity Recognition, noun chunk extraction, and RAKE keyword extraction) to extract an initial set of keywords from a standalone (but related) public dataset. We then sequentially update this set using an LLM. In each iteration, we present the LLM with the current set of keywords and each new conversation, prompting it to output words_to_remove and words_to_add. This sequential refinement continues until the keyword set stabilizes or reaches a desired size; see detailed prompt in §G.6.

**Combined Public-Private Selection (KwSet-Hybrid).** In this hybrid approach, we leverage KwSet-Public while maintaining privacy guarantees. We provide the LLM with KwSet-Public along with several private conversations, asking it to output relevant keywords only when necessary. We then apply partition selection over these selected keywords to ensure DP. We propose this approach to address potential distribution shifts between the public conversations used in constructing KwSet-Public and the private conversations we aim to analyze. This hybrid method allows for adaptation to new topics or terms in the private dataset while still benefiting from the high-quality foundation of the public keyword set. See detailed prompt in §G.7.

## 5 Evaluations

Our evaluation strategy adopts a multifaceted approach to assess the privacy-utility tradeoff in our DP pipeline. We compare summaries generated by our private URANIA pipeline against those from the (non-private) SIMPLE-CLIO on identical conversation sets, using the latter as a reasonable proxy for ground truth.

We employ both non-LLM methods (comparing key phrases, n-grams, topics, and embedding similarities) and LLM-based evaluations (comparative quality assessments and independent scoring) to comprehensively measure how well our privacy-enhanced summaries preserve the essential information captured in the public summaries. This approach allows us to quantify both the utility preservation and information retention of our DP pipeline while maintaining stronger formal privacy guarantees than the original framework. Additionally, we evaluate privacy protection through AUC score comparisons between the public and private pipelines, providing a more complete picture of the privacy-utility tradeoff in our system, which will be presented in the later sections (See §F).

### 5.1 Evaluation Methodology

Our evaluation methodology involves running both the public and private pipelines on the same conversation dataset, generating two sets of summaries: public (non-DP) and private (DP). We then compare these summaries using both automated metrics and LLM-based evaluations.

**Automated Evaluation.** We employ three automated approaches to measure the similarity between private and public summaries:

1. **Lexical Content Analysis:** We extract key phrases, noun chunks, and TF-IDF keywords from both sets of summaries and compute similarity metrics with Jaccard similarity between these feature sets.

2. **N-gram and Topic Analysis:** We analyze tokens and 2-grams extracted from both sets of summaries to assess content preservation at different granularities. Additionally, we use BERTopic (Grootendorst, 2022) to extract topics from both sets of summaries and measure their overlap.

3. **Embedding Space Proximity:** For each private summary, we compute its distance to the nearest public summary in the embedding space using the SentenceTransformer model with the all-mpnet-base-v2 embedding. This measures how well the private summaries preserve the semantic content of their public counterparts.

**LLM-based Evaluation.** We complement automated metrics with LLM-based evaluations that assess summary quality. In the prompt, we randomly chose the order of public and private summaries to avoid positional bias.

4. **Comparative Ranking:** We randomly sample conversations along with their corresponding private and public summaries. An LLM evaluator rates which summary is better on a scale from 1-5, where 1 indicates the private summary is clearly better and 5 indicates the public summary is clearly better.

5. **Binary Preference:** For sampled conversations, an LLM evaluator makes a binary choice between private and public summaries, selecting which one better summarizes the original conversation.

Tables 1 and 2 present the results of our comprehensive evaluation, comparing the private and public summaries across various metrics and configurations. The results of lexical content, n-gram and topic analysis are presented in Table 1, and the embedding space proximity and LLM-based evaluation results can be found in Table 2.

Finally, we propose a simple *empirical privacy evaluation* method using a membership inference-style attack to conceptually demonstrate that URANIA is indeed more robust than SIMPLE-CLIO; details in §F.

## 5.2 Experimental Setup & Results

Table 1: Lexical content, n-gram, and topic similarity Between Private and Public Summaries.

| Configuration | Key Phrases | Noun Chunks | TF-IDF | Tokens | 2-Grams | Topics |
|---|---|---|---|---|---|---|
| Very Low Privacy ($\varepsilon_c$=10.0, $\varepsilon_{hist}$=5.0) | 0.028 | 0.026 | 0.96 | 0.133 | 0.100 | 0.723 |
| Low Privacy ($\varepsilon_c$=8.0, $\varepsilon_{hist}$=4.0) | 0.026 | 0.025 | 0.96 | 0.130 | 0.096 | 0.461 |
| Medium Privacy ($\varepsilon_c$=4.0, $\varepsilon_{hist}$=2.0) | 0.027 | 0.023 | 0.96 | 0.113 | 0.077 | 0.271 |
| High Privacy ($\varepsilon_c$=2.0, $\varepsilon_{hist}$=1.0) | 0.035 | 0.021 | 0.94 | 0.063 | 0.033 | 0.078 |
| KwSet-LLM (Low Privacy) | 0.032 | 0.031 | 0.98 | 0.090 | 0.065 | 0.589 |
| KwSet-Public (Low Privacy) | 0.035 | 0.021 | 0.98 | 0.131 | 0.083 | 0.514 |
| KwSet-Hybrid (Low Privacy) | 0.030 | 0.022 | 0.96 | 0.145 | 0.092 | 0.446 |
| No Privacy ($\varepsilon = \infty$) | 0.024 | 0.027 | 0.98 | 0.139 | 0.112 | 0.907 |

Table 2: Embedding space proximity between and comparative LLM evaluation of private and public summaries.

| Configuration | Mean Cosine Similarity | Median | Comparative Ranking (1–5, lower is better) | Binary Preference (% DP preferred) |
|---|---|---|---|---|
| Very Low Privacy ($\varepsilon_c$=10.0, $\varepsilon_{hist}$=5.0) | 0.750 | 0.745 | 2.56 | 62 |
| Low Privacy ($\varepsilon_c$=8.0, $\varepsilon_{hist}$=4.0) | 0.749 | 0.744 | 2.40 | 65 |
| Medium Privacy ($\varepsilon_c$=4.0, $\varepsilon_{hist}$=2.0) | 0.748 | 0.742 | 2.39 | 69 |
| High Privacy ($\varepsilon_c$=2.0, $\varepsilon_{hist}$=1.0) | 0.774 | 0.773 | 2.50 | 64 |
| KwSet-LLM (Low Privacy) | 0.747 | 0.744 | 2.37 | 68 |
| KwSet-Public (Low Privacy) | 0.735 | 0.728 | 2.64 | 69 |
| KwSet-Hybrid (Low Privacy) | 0.739 | 0.732 | 2.55 | 70 |
| No Privacy ($\varepsilon = \infty$) | 0.759 | 0.754 | - | - |

We analyze the conversations from the popular public datasets `LMSYS-1M-Chat` (Zheng et al., 2023). Specifically we use `gemini-2.0-flash-001` language model. In constructing KwSet-Public, we use the public `WildChat` dataset (Zhao et al., 2024). In particular, we set the cluster size privacy parameter $\varepsilon_{size} = 1$ for cluster size verification and implement a 1-DP keyword set generation.

We evaluate our approach across different privacy parameter settings and keyword set configurations. We consider varying values of $\varepsilon_c$ and $\varepsilon_{hist}$, while keeping the keyword set to be generated using KwSet-TFIDF. We also fix $\varepsilon_c = 8.0$ and $\varepsilon_{hist} = 4.0$ and evaluate with different choices of keyword sets. Note that URANIA with $\varepsilon = \infty$ is still distinct from

SIMPLE-CLIO because the former generates cluster summaries via keywords, whereas the latter does so directly from individual summaries.

### 5.3 Discussion

The evaluation results reveal several key insights about the performance of our framework.

**Privacy-Utility Trade-off.** As expected, we observe a trade-off between privacy protection and summary quality. Higher privacy guarantees (lower $\varepsilon$ values) generally result in lower similarity to public summaries, particularly evident in the declining topic similarity scores in Table 1, where coverage drops from 0.723 at very low privacy to just 0.078 at high privacy. However, the degradation is not uniform across all metrics, suggesting that some aspects of summary quality are more robust to privacy noise than others.

**Impact of Keyword Set Selection.** The choice of keyword set significantly influences the quality of private summaries. Table 1 demonstrates that different keyword sets yield varying performance despite identical privacy parameters. For example, KwSet-LLM has better topic coverage compared to KwSet-TFIDF at the same level of privacy. This suggests that carefully curated keyword sets can partially mitigate the utility loss from DP.

**Topic Coverage and Preservation.** Following CLIO's evaluation approach, we treat public summaries as ground truth and measure the percentage of topics successfully captured by our DP method. Our private approach demonstrates reasonable topic coverage at lower privacy levels (0.723 at $\varepsilon_c = 10.0$), but this metric proves highly sensitive to increased privacy protection, precipitously declining to 0.078 at high privacy ($\varepsilon_c = 2.0$).

Examining the transition from very low privacy ($\varepsilon_c = 10.0$) to low privacy ($\varepsilon_c = 8.0$) in Table 1, while the number of private summaries decreases modestly from approximately 3,700 to 3,300 (an 11% reduction), topic coverage drops more dramatically from 0.723 to 0.461 (a 36% reduction). This disproportionate decline suggests that the DP clustering algorithm may systematically exclude certain conversation types even at this early privacy transition, though our current evaluation cannot determine which specific topics are affected. This sharp threshold effect raises important research questions about how different topic types respond to privacy constraints and whether alternative DP approaches might exhibit more gradual degradation.

A primary contributing factor to this performance degradation is our DP-KMEANS implementation. As the privacy budget tightens, the algorithm generates fewer viable cluster centers, with many centers positioned suboptimally relative to actual data points. Consequently, some centers fail to attract any conversation assignments, further compromising topic coverage at higher privacy levels.

The clustering performance illustrates this progressive decline quantitatively: the number of final clusters decreases from approximately 3,700 at very low privacy to just 300 at high privacy configurations. While this trend underscores the cascading effect of DP on clustering quality, it also raises an intriguing research question: Can alternative DP-clustering methods or novel implementations mitigate these limitations? Although such performance degradation appears somewhat inherent to DP techniques, which fundamentally trade granularity for privacy protection, exploring alternative algorithmic approaches may reveal strategies to better preserve minority representations while maintaining robust privacy guarantees.

**Semantic Preservation.** The embedding-based evaluation in Table 2 shows consistently high cosine similarity scores across all configurations (0.73–0.77), indicating that private summaries generally maintain the semantic essence of their public counterparts, even at higher privacy levels. This is particularly notable because it suggests that overall meaning is preserved even when specific lexical content differs.

**LLM Evaluation Insights.** Table 2 shows that LLM evaluators sometimes prefer private summaries over public ones, with comparative ranking scores generally below 3 (where 3 would indicate no preference). The binary preference results further confirm this trend, with 62–70% of evaluations favoring the DP-generated summaries. This suggests that the

constraints imposed by our DP approach (limiting to predetermined keywords, focusing on most frequent themes) can occasionally produce more concise and focused summaries than the unconstrained public approach.

**Independent Scoring Results.** We have an LLM evaluator independently score private and public summaries on a scale from 1-5 (where 1 means very poor and 5 means excellent) based on how well they summarize the original conversations. The private summaries are slightly better than public summaries, but both achieve very low scores on average ($<1.4$). This suggests that there might be significant room for improvement in the overall quality of conversation summarization, regardless of privacy considerations, and highlights the challenging nature of the summarization task itself.

These findings highlight both the capabilities and limitations of DP text summarization. Our framework demonstrates that it is possible to generate meaningful summaries while providing formal privacy guarantees, but practitioners should carefully consider the privacy-utility trade-off when configuring the system for real-world applications. However, qualitative analysis of specific examples (see §D) reveals important limitations in our private summaries that may not be captured by automated evaluation metrics.

## 6 Discussion and Future Work

Our work demonstrates that meaningful privacy guarantees can be achieved while maintaining useful conversation summarization capabilities. We identify several future directions:

**Stronger Privacy Attacks.** While we formalized a specific empirical privacy evaluation, developing stronger attacks against non-DP summarization systems remains an important direction. The privacy vulnerability of clustering-based algorithms is relatively unexplored compared to other machine learning paradigms. More sophisticated attacks would provide valuable insights into the precise privacy risks of non-DP systems and better quantify the benefits of DP approaches.

**Online Learning and Adaptation.** An important extension would be adapting our framework to online settings where new conversations continuously arise. This presents challenges in maintaining privacy guarantees while incorporating new data, evolving keyword sets to capture emerging topics, and efficiently updating cluster structures. This remains an open problem for real-world applications of private conversation summarization.

**Utility Improvements.** Further improvements could narrow the quality gap between private and non-private summarization approaches. These include exploring alternate privacy mechanisms with better utility-privacy trade-offs, developing more sophisticated keyword selection methods, and refining summarization prompts to generate more informative cluster summaries.

**User-level DP.** Our current pipeline exclusively addresses DP at the record-level, where privacy guarantees are provided when only a single conversation is modified. However, in practical scenarios, individual users typically contribute multiple conversations with large language models (LLMs). This motivates an important open research problem: extending the privacy notion to user-level DP, wherein the privacy guarantees hold when all conversations from a single user are removed.

**Broader Privacy Landscape.** While our work focuses on formal DP guarantees for released summaries, we acknowledge that comprehensive privacy protection requires addressing multiple risk vectors. Our approach provides one layer of protection against information leakage through published analyses, but does not address other important concerns such as centralized data collection risks, third-party model processing vulnerabilities, or inference attacks on the underlying conversation data. Future work should explore how formal DP guarantees can be integrated with other privacy-preserving techniques (such as federated learning, secure multi-party computation, or local DP) to create more comprehensive privacy frameworks for conversation analysis systems.

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

# A  Formal Details of DP Tools

In this section we recall basic properties of DP as well as formally describe the tools of private partition and private histogram release as alluded to in §2.

**Proposition A.1** (Properties of DP)**.**

- **[Parallel Composition]** *If $\mathcal{M} : \Omega^* \to \mathcal{R}$ satisfies $(\varepsilon, \delta)$-DP, then for any disjoint collection of subsets $\Omega_1, \ldots, \Omega_r \subseteq \Omega$, given a dataset $D \subseteq \Omega$, the mechanism that returns $(\mathcal{M}(D \cap \Omega_1), \ldots, \mathcal{M}(D \cap \Omega_r))$ also satisfies $(\varepsilon, \delta)$-DP.*

- **[Basic Composition]** *If mechanism $\mathcal{M}_1$ with output space $Y$ satisfies $(\varepsilon_1, \delta_1)$-DP and the mechanism $\mathcal{M}_{2,y}$ satisfies $(\varepsilon_2, \delta_2)$-DP for all $y \in Y$. Then the mechanism $\mathcal{M}$ that maps $x$ to $(y_1 \sim \mathcal{M}_1(x), y_2 \sim \mathcal{M}_{2,y_1}(x))$ satisfies $(\varepsilon_1 + \varepsilon_2, \delta_1 + \delta_2)$-DP.*

- **[Postprocessing]** *If $\mathcal{M}$ with output space $Y$ satisfies $(\varepsilon, \delta)$-DP, then for any function $f : Y \to Z$, the mechanism $\mathcal{M}'$ that maps $x \mapsto f(M(x))$ satisfies $(\varepsilon, \delta)$-DP.*

The (truncated) discrete Laplace distribution $\mathsf{DLap}_\tau(\varepsilon)$ is the distribution on $\mathbb{Z}$ such that

$$\Pr_{X \sim \mathsf{DLap}_\tau(\varepsilon)}[X = x] = \begin{cases} \beta_{\varepsilon,\tau} \cdot e^{-\varepsilon |x|} & \text{for } z \in [-\tau, \tau] \\ 0 & \text{otherwise} \end{cases}, \qquad \text{where } \beta_{\varepsilon,\tau} := \frac{1 - e^{-\varepsilon}}{1 + e^{-\varepsilon} - 2e^{-\varepsilon(\tau+1)}}$$

The case of $\tau = \infty$ is simply the discrete Laplace distribution $\mathsf{DLap}(\varepsilon)$.

## A.1  Private Histogram Release

The mechanism $\mathsf{PHR}_{k,\varepsilon}(h_1, \ldots, h_n)$ simply returns a noisy histogram obtained by adding discrete Laplace noise $\mathsf{DLap}(\varepsilon/k)$ to each value of the true histogram.

---

**Algorithm 2** Private Histogram Release $\mathsf{PHR}_{k,\varepsilon}(x_1, \ldots, x_n)$

---

**Parameters:** $\varepsilon > 0$ and $k \in \mathbb{N}$.
**Input:** $h_1, \ldots, h_n \subseteq \mathcal{B}$ with $|h_i| \leq k$.
**Output:** A histogram $\bar{h} \in \mathbb{Z}^{\mathcal{B}}$

$h^* \leftarrow \sum_i h_i$ ;                    // $h_i$ interpreted as indicator vector in $\{0,1\}^{\mathcal{B}}$.
**for** $b \in \mathcal{B}$ **do**
$\quad$ Let $\zeta_b \sim \mathsf{DLap}(\varepsilon/k)$
$\quad$ $\bar{h}_b \leftarrow h^*_b + \zeta_b$
**return** $\bar{h}$

---

## A.2  Private Partition Selection

The mechanism $\mathsf{PPS}_{k,\varepsilon,\delta}(h_1, \ldots, h_n)$ returns a subset of elements for which the noisy count, after adding truncated discrete Laplace noise $\mathsf{DLap}_\tau(\varepsilon/k)$ is more than $\tau$. The result for $k = 1$ was provided by Desfontaines et al. (2022) and the result for general $k$ can be derived using the *group privacy* property of DP (see Lemma 2.2 in Vadhan (2017)).

---

**Algorithm 3** Private Partition Selection $\text{PPS}_{k,\varepsilon,\delta}(x_1,\ldots,x_n)$

---

**Parameters:** $k \in \mathbb{N}$, $\varepsilon > 0$ and $\delta \in [0,1]$.
**Input:** $h_1,\ldots,h_n \subseteq \Omega$ with $|h_i| \leq k$.
**Output:** A set $R \subseteq \bigcup_i h_i$.

$\varepsilon', \delta' \leftarrow \varepsilon \cdot \frac{1}{k}, \delta \cdot \frac{e^{\varepsilon/k}-1}{e^{\varepsilon}-1}$
$\tau \leftarrow \lceil \frac{1}{\varepsilon'} \log(\frac{e^{\varepsilon'}+2\delta'-1}{(e^{\varepsilon'}+1)\delta'}) \rceil$
$R \leftarrow \varnothing$
**for** $\omega \in \bigcup_i h_i$ **do**
  $c_\omega \leftarrow |\{i : h_i \ni \omega\}|$
  $\zeta_\omega \sim \text{DLap}_\tau(\varepsilon)$
  **if** $c_\omega + \zeta_\omega > \tau$ **then**
    $R \leftarrow R \cup \{\omega\}$
**return** $R$

---

# B   Description of SIMPLE-CLIO

We formally describe the SIMPLE-CLIO algorithm in Algorithm 4 complementing the description in §3.

---

**Algorithm 4** SIMPLE-CLIO$_k$ : Non-Differentially Private Text Corpora Summarization

---

**Parameters:** $k$ : parameter for KMeans clustering
**Input:** Input dataset $\boldsymbol{x} = (x_1, x_2, \ldots, x_n) \in \mathcal{X}^n$
**Output:** High Level Summaries $\boldsymbol{s} = (s_1, \ldots, s_k) \in \mathcal{S}^k$

```
// Step 1: Extract Embeddings from Conversations
```
$e \leftarrow \{\text{ExtractEmbeddings}(x_i) : i \in [n]\}$
```
// Step 2: Clustering (K-means)
```
$\boldsymbol{c}, \boldsymbol{C} \leftarrow \text{KMEANS}_k(e)$ ;                              `// c are centers, C are assignments`
```
// Step 3: Summarize each Cluster
```
$\boldsymbol{s} \leftarrow \varnothing$ ;                              `// Initialize summaries set`
**for** $j \leftarrow 1$ **to** $k$ **do**
  $\textbf{samples}_j \leftarrow \text{SampleConversations}(c_j, C_j, \boldsymbol{x})$ ;   `// Draw representative samples based on`
    `the center and cluster`
  $s_j \leftarrow \text{LLM-Summarize}(\textbf{samples}_j)$ ;                `// Generate summary using LLM`
  $\boldsymbol{s} \leftarrow \boldsymbol{s} \cup \{s_j\}$
**return** $\boldsymbol{s}$

---

The sub-routines used in the above algorithm operate as follows:

- ExtractEmbeddings($x$) uses an LLM to create a "summary" for $x$ (see prompt in §G.1), and thereafter uses an embedding model that maps the summary to a real-valued vector. Specifically, we use `all-mpnet-base-v2` embedding model (Reimers & Gurevych, 2019; 2022).

- KMEANS$_k(e)$ applies the $k$-means algorithm that partitions the vectors $e$ into $k$ clusters that minimizes the $k$-means objective.

- SampleConversations draws a small (e.g., 10) number of random conversations from the specified cluster.

- LLM-Summarize uses an LLM to generate a summary associated to the cluster using the sampled conversations from the cluster (see prompt in §G.2).

## C Keyword Selection Algorithms

We formally describe the methods for extract keywords from documents within each cluster, complementing the description in §4.1. In these methods, we apply the extraction on the concatenation of the various facets of the inputs $x_i$'s. However, for simplicity, we write the pseudocode assuming they are applied on the raw inputs.

**TF-IDF-Based Selection with Partition Selection (KwSet-TFIDF).** This method proceeds by constructing a token frequency matrix $\mathbf{TF} \in \mathbb{Z}^{n \times |V|}$ where $\mathbf{TF}_{i,j}$ is the number of times the token $j$ appears in input $x_i$. We normalize the token frequency matrix such that each column has $\ell_1$ norm at most $w_{\max}$.

Additionally, we construct a *document frequency* vector $\mathbf{df} \in \mathbb{Z}^{|V|}$, where $\mathbf{df}_j$ is the number of times the token $j$ appears across all documents. This is privatized to obtain $\widetilde{\mathbf{df}}$ by adding discrete Laplace noise. And similarly, the number of documents is estimated as $\tilde{n}$ by adding discrete Laplace noise to $n$.

The *inverse document frequency* $\mathbf{idf}$ is set as $\log(\tilde{n}/\widetilde{\mathbf{df}})$. And finally, the $\mathbf{TFIDF}$ matrix is constructed by multiplying each column of the $\mathbf{TF}$ matrix with the $\mathbf{idf}$ vector.

---

**Algorithm 5** Differentially Private TF-IDF Keyword Extraction (KwSet-TFIDF)

---

**Parameters:** Maximum keywords per conversation $k$, maximum document weight $w_{\max}$, maximum total keywords $K$.
**Parameters:** Privacy parameters $\varepsilon_{\text{idf}}, \varepsilon_{\text{sel}} > 0, \delta_{\text{sel}} \in [0, 1]$.
**Input:** Input dataset $\boldsymbol{x} = (x_1, x_2, \ldots, x_n)$ of structured facets
**Output:** Differentially private keyword set $\mathcal{K}$

```
// Step 1: Build TF matrix and apply L1 clipping
```
$\mathbf{TF}, \mathcal{V} \leftarrow \text{CountVectorizer}(\boldsymbol{x})$ ;                    `// Build term frequency matrix and token list`
**for** $i \leftarrow 1$ **to** $n$ **do**
    **if** $\|\mathbf{TF}_{i,:}\|_1 > w_{\max}$ **then**
        $\mathbf{TF}_{i,:} \leftarrow \mathbf{TF}_{i,:} \cdot \frac{w_{\max}}{\|\mathbf{TF}_{i,:}\|_1}$ ;                    `// Clip document weights`
```
// Step 3: Compute noisy document frequencies
```
$\mathbf{df} \leftarrow \sum_{i=1}^{n} \mathbf{TF}_{i,j}$ for $j \in |\mathcal{V}|$ ;                    `// Compute document frequencies`
$\sigma \leftarrow \frac{\varepsilon_{\text{idf}}}{w_{\max}+1}$ ;                    `// Noise scale`
$\widetilde{\mathbf{df}} \leftarrow \max(\mathbf{df} + \text{DLaplace}(\sigma), 1)$ ;                    `// Add DP noise`
$\tilde{n} \leftarrow n + \text{DLaplace}(\sigma)$ ;                    `// Noisy document count`
```
// Step 4: Compute DP TF-IDF and extract keywords
```
$\mathbf{idf}^{DP} \leftarrow \log(\tilde{n}/\widetilde{\mathbf{df}})$ ;                    `// Compute DP IDF`
$\mathbf{TFIDF}^{DP} \leftarrow \mathbf{TF} \odot \mathbf{idf}^{DP}$ ;                    `// Compute DP TF-IDF`
$h_i \leftarrow \text{top } k \text{ tokens from } \mathbf{TF}_{i,:}$ for each $i \in [n]$ ;                    `// Extract top k keywords per conversation`
```
// Step 5: Apply DP selection mechanism
```
$\mathcal{K} \leftarrow \text{PPS}_{k, \varepsilon_{\text{sel}}, \delta_{\text{sel}}}(h_1, \ldots, h_n)$ ;                    `// Select final keywords with DP`
```
// Step 6: Limit to maximum keywords (optional)
```
**if** $|\mathcal{K}| > K$ **then**
    $\mathcal{K} \leftarrow \text{TopK}(\mathcal{K}, K, \text{by original counts})$
```
// Step 7: LLM refinement
```
$\mathcal{K} \leftarrow \text{LLMRefineKeywords}(\mathcal{K}, K)$ ;                    `// See prompt in §G.5`
**return** $\mathcal{K}$

---

---

**Algorithm 6** DP Selection from Facet Keywords with LLM Refinement (KwSet-LLM)

---

**Parameters:** Maximum keywords per conversation $m$, maximum total keywords $K$.
**Parameters:** Privacy parameters $\varepsilon_{\text{sel}} > 0$, $\delta_{\text{sel}} \in (0, 1]$.
**Input:** Input dataset $\boldsymbol{x} = (x_1, x_2, \ldots, x_n) \in \mathcal{X}^n$
**Output:** Differentially private keyword set $\mathcal{K}_{\text{refined}}$

```
// Step 1: Extract a set of at most m keywords h_i for each x_i.
```
**for** $i = 1, \ldots, n$ **do**
  $\quad h_i \leftarrow \text{GetKeywords}(x_i)$ ;                    `// See prompt template in §G.5.`
```
// Step 2: Apply DP selection mechanism
```
$\mathcal{K} \leftarrow \text{PPS}_{k, \varepsilon_{\text{sel}}, \delta_{\text{sel}}}(h_1, \ldots, h_n)$
```
// Step 3: LLM refinement
```
**while** $|\mathcal{K}| > K$ **do**
  $\quad \mathcal{K}_{\text{refined}} \leftarrow \text{LLMQuery}(\text{RefinePrompt}(\mathcal{K}))$ ;          `// LLM refinement (see §G.5)`
**return** $\mathcal{K}_{refined}$

---

**Algorithm 7** LLM-guided Sequential Keyword Refinement (KwSet-Public)

---

**Parameters:** Initial batch size $n_0$, sequential batch size $b$, maximum initial keywords $K_0$.
**Input:** Input dataset $\mathbf{x} = (x_1, x_2, \ldots, x_n)$ of structured facets
**Output:** Refined keyword set $\mathcal{K}^{pub}$

```
// Step 1: Extract text and split dataset
```
$\boldsymbol{x}_{\text{init}} \leftarrow \boldsymbol{x}[1 : n_0], \boldsymbol{x}_{\text{seq}} \leftarrow \boldsymbol{x}[n_0 + 1 : n]$ ;            `// Split into initial and sequential sets`
```
// Step 2: Extract and refine initial keywords
```
$\mathbf{K}_{raw} \leftarrow \bigcup_{x \in \boldsymbol{x}_{init}} \{\text{NER}(x) \cup \text{NounChunks}(x) \cup \text{RAKE}(x)\}$ ;        `// Multi-method extraction`
$\mathcal{K}^{pub} \leftarrow \text{LLMRefine}(\mathbf{K}_{raw})$ ;                `// Remove redundancy, merge concepts (see §G.6)`
$\mathcal{K}^{pub} \leftarrow \text{RandomSample}(\mathcal{K}^{pub}, \min(|\mathcal{K}^{pub}|, K_0))$ ;                    `// Limit initial set`
```
// Step 3: Sequential updating with LLM feedback
```
**for** $i \leftarrow 1$ **to** $\lceil |\boldsymbol{x}_{seq}|/b \rceil$ **do**
  $\quad \text{batch\_text} \leftarrow \text{Join}(\boldsymbol{x}_{seq}[(i-1)b + 1 : ib])$ ;                    `// Merge batch conversations`
  $\quad (\textbf{remove}, \textbf{append}) \leftarrow \text{LLMUpdate}(\mathcal{K}^{pub}, \text{batch\_text})$ ;   `// Get update suggestions (see §G.6)`
  $\quad \mathcal{K}^{pub} \leftarrow (\mathcal{K}^{pub} \setminus \textbf{remove}) \cup \textbf{append}$ ;                    `// Apply updates`
**return** $\mathcal{K}^{pub}$

---

---

**Algorithm 8** Combined Public-Private DP Keyword Selection (KwSet-Hybrid)

---

**Parameters:** Maximum number of keywords $k$ per conversation.
**Parameters:** Privacy parameters $\epsilon_{\text{hyb}} > 0$, $\delta_{\text{hyb}} \in (0, 1]$.

**Input:** Public keyword set $\mathcal{K}^{pub}$, input dataset $x = (x_1, x_2, \ldots, x_n)$
**Output:** Hybrid DP keyword set $\mathcal{K}^{hyb}$

// Step 1: LLM-guided candidate extraction in batches
**for** $i = 1$ *to* $n$ **do**
$\quad h_i \leftarrow \textsf{GetNovelKeywords}(\mathcal{K}^{pub}, x_i)$ ;                    // Using prompt; see §G.7
// Step 2: Partition selection to select keywords corresponding to private conversations
$\mathcal{K}_{\text{private}} \leftarrow \textsf{PPS}_{k,\varepsilon,\delta}(h_1, \ldots, h_n)$
**return** $\mathcal{K}_{private} \cup \mathcal{K}_{pub}$

---

# D    Summary Comparison Examples

To better understand the trade-offs between public and private summaries, we present several illustrative examples that highlight the key differences in specificity and contextual relevance.

Table 3: Comparison of Public vs Private Summary Examples

| |
| --- |
| **CONVERSATION:** |
| The user is asking how to extract the last character of a file in BASH and convert it to hexadecimal. |
| **SUMMARIES:** |
|     **Public:** Scripting and Command-Line Task Automation Requests |
|     **Private:** Software Development, Version Control, and Educational Resources |
| **CONVERSATION:** |
| The assistant explains the key differences between reinforcement learning and unsupervised learning, focusing on feedback and goals. |
| **SUMMARIES:** |
|     **Public:** Deep Learning, Training, and Applications Explanation |
|     **Private:** AI Model Training, Deployment, and Management Considerations |
| **CONVERSATION:** |
| The user is asking if the assistant speaks Russian and the assistant confirms and offers help. |
| **SUMMARIES:** |
|     **Public:** Russian Language Support and Communication |
|     **Private:** AI-Powered Applications and Open Source Technology |

As these examples demonstrate, public summaries tend to be more specific and directly relevant to the actual conversation content, while private summaries are often broader and sometimes miss key contextual details. This limitation stems from the privacy constraints that prevent fine-grained keyword extraction—for instance, in the language example, specific terms like "Russian" may not appear in our predetermined keyword sets, leading to generic categorizations that fail to capture the conversation's essence.

These patterns suggest that while LLM evaluators may prefer the broader categorizations produced by our private pipeline for their apparent comprehensiveness, human evaluators might better discern the loss of specificity and contextual relevance. The trade-offs observed in these examples highlight the necessity of incorporating human-based evaluation in future work to provide a more nuanced assessment of summary quality, as human judgment may be more sensitive to the subtle but important differences in semantic accuracy and practical utility that automated metrics might overlook.

# E    Additional Related Work

**Analysis of LLM conversations** Our work builds on real-world conversations with LLM chatbots from recent open-source datasets (Zheng et al., 2023; Zhao et al., 2024). Prior work has analyzed such conversations to extract structure and insight. For instance, Zhao et al. (2024) define a set of common conversation categories and prompt LLMs to classify conversations accordingly. In contrast, Zheng et al. (2023) and Tamkin et al. (2024) adopt more flexible approaches that do not rely on predefined categories. Zheng et al. (2023) cluster conversations in the embedding space and then use LLMs to summarize each cluster's main theme. Tamkin et al. (2024) build on this by first summarizing individual conversations into concise summaries and then applying hierarchical clustering. While these works provide valuable frameworks for understanding LLM behavior, none offer formal privacy guarantees. Our work addresses this gap by introducing the first DP conversation summarization pipeline.

**Clustering.** Both Tamkin et al. (2024) and our work employ a clustering step to group conversations with similar embeddings together. We use $k$-means, which is one of the most popular formulations of clustering; see e.g. Ikotun et al. (2023) for a comprehensive survey. For non-DP $k$-means, we use the classic algorithm of Lloyd (1982). For DP $k$-means, several algorithms have been proposed, starting with Blum et al. (2005) who devised a DP version of Lloyd's algorithm. This was followed up by a series of works improving different algorithmic guarantees (e.g. Nissim et al. (2007); Gupta et al. (2010); Balcan et al. (2017); Ghazi et al. (2020)), culminating in the algorithm of Chang et al. (2021) which not only gives a nearly optimal (theoretical) approximation ratio but is also practical. We ended up using the open-source library Chang & Kamath (2021) based on this paper. We remark that there are also more recent works which provide practical DP clustering algorithms, e.g., Tsfadia et al. (2022); nevertheless, we are unaware of any open-source library based on these papers.

**Membership inference attacks (MIA)** Our empirical privacy evaluation (§F) is in spirit similar to MIA, which aims to infer whether specific data points were included in the training data of a given model or algorithm, based on its outputs. While various MIA techniques have been proposed, they primarily target machine learning models (Shokri et al., 2017; Yeom et al., 2018; Sablayrolles et al., 2019; Carlini et al., 2022). In this work, we design an empirical privacy evaluation method tailored to attack the LLM conversation summarization pipeline.

**Keyword generation with LLMs** Our keyword set generation method (§4.1) is related to recent work on extracting keywords from text corpora using LLMs (Grootendorst, 2020; Maragheh et al., 2023; Wang et al., 2024). These approaches typically fine-tune or prompt an LLM to extract keywords. However, such pipelines are not applicable in a DP setting, as documents may contain unique keywords that could leak sensitive information. In this work, we propose keyword generation methods that only output keywords shared across multiple conversations, allowing us to provide formal DP guarantees.

# F   Empirical Privacy Leakage Evaluation

To empirically evaluate privacy protection, we conducted a simple experiment to compare the privacy leakage of our private and the public pipelines. Specifically, we created a synthetic dataset of 100 conversations: 1 sensitive conversation on health/medical topics and 99 non-sensitive conversations on general topics (food, travel, homework help and health). We ran both pipelines on this dataset, then quantify the privacy leakage by measuring the maximum embedding similarity between the sensitive conversation and the generated summaries. We consider a simple thresholding-based detector of the sensitive conversation, and measure the AUC under different thresholds under 200 runs where half of the runs include the sensitive conversation.

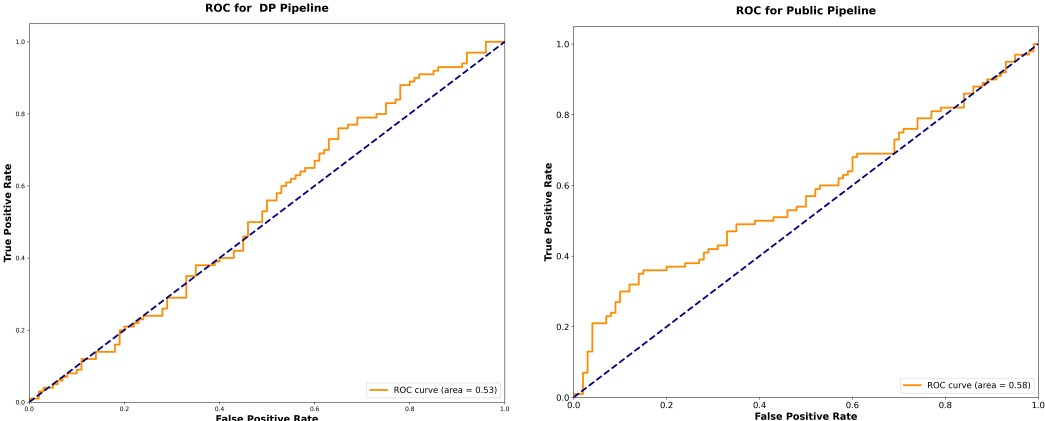

(a) Empirical Privacy Leakage Measurement against DP pipeline ($\varepsilon = 21$, AUC = 0.53)

(b) Empirical Privacy Leakage Measurement against public pipeline (AUC = 0.58)

Figure 2: ROC curves for Empirical Privacy Leakage. The DP pipeline (a) shows performance equivalent to random guessing (AUC $\approx$ 0.53), while the public pipeline (b) is more vulnerable with an AUC of 0.58.

Figure 2 shows the results. Our DP approach achieved an AUC of only 0.53, effectively equivalent to random guessing (0.5), demonstrating strong privacy protection. In contrast, the non-private pipeline showed greater vulnerability with an AUC of 0.58, indicating that embedding similarity can leak information of the sensitive conversations when privacy mechanisms are absent.

It is worth noting that our experiment used casually generated synthetic conversations rather than carefully crafted adversarial examples. A more sophisticated experiment might potentially achieve better performance against either pipeline. However, even with this simple approach, we can observe a clear difference in privacy protection between the DP and non-DP methods.

These results provide empirical evidence that our DP approach offers practical protection against such privacy leakage while maintaining useful summarization capabilities, even when using a relatively high privacy budget ($\varepsilon = 21$) on a relatively small dataset (of no more than 100 conversations).

## G LLM Prompts

We provide some example prompts we used in the pipeline.

### G.1 Extracting Embeddings from Records

Below, we provide the prompt used in the ExtractEmbeddings method in Algorithm 4 and Algorithm 1. The input record $x$ is substituted in place of $\langle x \rangle$ in the prompt. Note that we apply the embedding function specifically to the 'Summary' portion of the LLM response obtained using this prompt, rather than to the entire response.

---

**Prompt for ExtractEmbeddings($x$)**

```
You are an advanced AI assistant that processes conversations and extracts meaningful
facets for downstream embedding and clustering tasks. For the conversation provided,
return a structured JSON object with the following:
- Topics: The overarching topics discussed in the conversation.
- Subtopics: Specific details or points under each topic.
- Intent:  The main intent of the conversation (e.g., seeking help, sharing an
opinion).
- Entities: Important names, organizations, or proper nouns discussed.
- Keywords:  The most important words/phrases that could help differentiate the
conversation.
- Summary: A concise, one-sentence description summarizing the conversation, up to
20 words.

Ignore the conversation if it is not in English.
Output the result as a valid JSON object with no extra text.

### Example:
User: I'm having a hard time preparing for job interviews. How do I answer behavioral
questions? I also get nervous. Any tips on how to stay calm during the process?

Expected JSON Output:
{
  'Topics': ['Job Interviews', 'Behavioral Questions', 'Stress Management'],
  'Subtopics': ['Handling difficult questions', 'Managing anxiety during interviews',
'Tips for preparation'],
  'Intent': 'Seeking advice',
  'Entities': ['job interview', 'behavioral questions'],
  'Keywords': ['job preparation', 'nerves', 'behavioral interview tips'],
  'Summary': 'The user is seeking advice on preparing for behavioral questions and
managing stress during job interviews.',
}

### Now, analyze the following conversation and generate the output in the same format:

User: ⟨x⟩

Return the structured output in JSON format.
```

### G.2 LLM-Summarize **in SIMPLE-CLIO (Algorithm 4)**

We provide the prompt used for LLM-Summarize method in SIMPLE-CLIO. In both CLIO and our proposed pipeline implementations, we select contrastive summaries from the nearest points located outside the designated cluster. For the sake of algorithmic clarity, we omitted this nuanced selection strategy from Algorithm 4 to maintain a simplified representation of the core approach.

---

**LLM-Summarize**

```
You are tasked with summarizing a group of related statements into a short, precise,
and accurate description and name. Your goal is to create a concise summary that
captures the essence of these statements and distinguishes them from other similar
groups of statements.

Summarize all the statements into a clear, precise, two-sentence description in the
past tense. Your summary should be specific to this group and distinguish it from
the contrastive answers of the other groups. After creating the summary, generate a
short name for the group of statements. This name should be at most ten words long
(perhaps less) and be specific but also reflective of most of the statements (rather
than reflecting only one or two).

Present your output in the following format:
<summary> [Insert your two-sentence summary here] </summary>
<name> [Insert your generated short name here] </name>

Below are the related statements:
<answers>
- ⟨summary 1 in within_cluster_summaries⟩
- ⟨summary 2 in within_cluster_summaries⟩
- ...
</answers>

For context, here are statements from nearby groups that are NOT part of the group
you're summarizing:
<contrastive_answers>
- ⟨summary 1 in contrastive_summaries⟩
- ⟨summary 2 in contrastive_summaries⟩
- ...
</contrastive_answers>

Do not elaborate beyond what you say in the tags. Remember to analyze both the
statements and the contrastive statements carefully to ensure your summary and name
accurately represent the specific group while distinguishing it from others.
```

---

### G.3 LLMSelectKeywords

The following prompts are used to extract keywords given the conversation and the KwSet.

---

**LLMSelectKeywords**

Below is a summary of a conversation. Based on this summary, select the most relevant
{m} keywords from the given set of keywords.

Summary: {summary}

Available Keywords: {keyword$_1$}, {keyword$_2$}, ...

Select up to {m} most relevant keywords from the set and return them as a Python list
(e.g., ['keyword1', 'keyword2', ...]).
If there is no relevant keyword, return the Python list ['NA']

---

### G.4 LLM-Summarize in Algorithm 1

We modified the prompt for Algorithm 4, and got the following prompt for LLM-Summarize
in Algorithm 1.

---

**LLM-Summarize**

You are a language model tasked with summarizing a group of related keywords and
example texts into a concise topic description.
Your goal is to generate:
1. A **concise topic name** ($\leq$10 words) that best describes the theme of these
keywords and examples.
2. A **brief, 2-sentence description** explaining what this cluster is about.

Please return your response in the following format:

<topic> [Insert topic name here] </topic>
<description> [Insert brief summary here] </description>

Below are the keywords associated with the cluster:
<keywords>
{keyword$_1$}, {keyword$_2$}, ...
</keywords>

Ensure the topic name is **specific**, descriptive, and meaningful.

---

### G.5 Prompts for KwSet-LLM and KwSet-TFIDF

We used LLM to choose keywords from conversations in the process of generating KwSet. We provide the prompt for generating KwSet-LLM for example.

---

**KwSet-LLM Generation (Before Refinement)**

```
You are a knowledgeable AI assistant tasked with generating a set of comprehensive
keywords that cover the topics, intents, and entities present in a dataset of human
conversations.

For the summary provided, generate a list of **3-5** keywords that reflect:
- The core topic(s) of the conversation
- Key entities or names mentioned
- Keywords useful for identifying conversation categories or topics

Keep the keywords concise and relevant.

Summary: {conversation_summary}

Use this JSON schema:
Response = {{"keywords": list[str]}}

Return only the JSON object without any explanations, decorations, or code blocks.
Only the raw JSON should be returned.
```

---

As a keyword set of smaller size leads to smaller utility loss due to privacy, we use an LLM to refine the keyword set (for both KwSet-TFIDF and KwSet-LLM) using the following prompt template.

---

**Refine KwSet**

```
You are an expert in topic summarization. We have extracted keywords from a large
collection of conversations. These keywords are used for summarizing the main topics
of the conversations based on sentence statistics. Your task is to refine this list
and select the most relevant {num} keywords that best capture the core topics.

Here is the extracted keyword list:

{json.dumps(keywords, indent=2)}

Please return exact {num} of the most relevant keywords in JSON format using the
following schema:

```json
{{
"keywords": ["keyword1", "keyword2", ..., "keyword{num}"]
}}
```
Do not exceed {num} keywords. If necessary, prioritize the most commonly occurring,
diverse, and representative words.
```

---

### G.6 Prompt for KwSet-Public

---

**Refine KwSet-Public**

```
You are refining a set of extracted key phrases for topic summarization.

Below is a list of candidate keywords extracted from conversations:
[', '.join(all_candidate_keywords)]

Your task:
- **Remove meaningless words** such as random numbers, dates, single letters, generic
words (e.g., "thing", "something", "stuff").
- **Delete overly specific entity names** (e.g., "John Doe", "xyz123", "April 5,
2023", "ID number").
- **Remove redundant words** that mean the same thing.
- **Merge similar concepts** (e.g., "AI fairness" and "bias in AI" should be unified).
- **Keep only the most informative, meaningful terms** that are useful for summarizing
topics.
- **Avoid vague or overly broad words** like "technology", "question", "people".
- **Ensure the final keywords are suitable for clustering and summarization**.

Return only the **refined keyword set** as a comma-separated list.
```

---

**Update KwSet-Public by adding and removing keywords**

```
You are an expert in topic summarization and keyword analysis. The current public
keyword set contains {len(existing_keywords)} words:
[{', '.join(existing_keywords))}]

Here is a batch of private conversations:
'{batch_text}'

Your task:
- Compare the batch of private conversations with the current keyword set.
- Identify keywords in the current set that are redundant or irrelevant.
- Identify new, meaningful keywords present in the private conversations that are
not in the current set.
- Output ONLY a complete, valid JSON object (with no extra text) with the following
structure:

{{
    'words_to_remove': [list of words to remove],
    'words_to_append': [list of words to add]
}}

Ensure your output starts with '{' and ends with '}' (i.e., the JSON object must be
complete, including the closing bracket). Keep your answer as concise as possible.
```

### G.7 Prompt for KwSet-Hybrid

---

**Refine KwSet-Hybrid**

```
You are an expert in topic analysis and keyword extraction.

The current public keyword set contains {len(public_keywords)} words:
[{', '.join(public_keywords)}]

Here is a batch of private conversations:
'{batch_text}'

Your task:
- Analyze the above conversations and identify any new, meaningful keywords that are
not already present in the public set.
- Omit any non-English conversations or keywords.
- Return ONLY a concise, comma-separated list of candidate keywords with no additional
explanation of at most 5 words.
```

---

### G.8 Prompt for Evaluation

As discussed before, we do some LLM-based evaluations. Now we provide the prompts for the binary preference.

---

**Prompt for Binary Preference**

```
You are an expert evaluator of text summarization systems. You will be given an
original text and two different summaries of that text. Your task is to evaluate
which summary better captures the key themes and content of the original text.

Original Text:
{text}

Summary A:
{summary_a}

Summary B:
{summary_b}

Please evaluate which summary better captures the key themes and content of the
original text. Consider factors such as:
- Accuracy of the information
- Coverage of important topics
- Clarity and coherence
- Relevance to the original text

Provide your reasoning, then end with a clear choice in the format: <choice>X</choice>
where X is either "A" for the first summary or "B" for the second summary.
```

---

# H Hierarchy and Visualization

## H.1 Hierarchical Organization

Table 4: Examples of Summaries and Cluster Levels

---

**SUMMARY:**

The conversation starts with greetings in Portuguese, inquiring about the other person's well-being.

**CLUSTERING:**

**Base:** AI Travel Assistant for Multilingual and Localized Recommendations

**Top:** AI Assistant Capabilities, Performance, and Applications

---

**SUMMARY:**

The user requested a Python program to create a SQLite database table named 'legacy' with specified fields.

**CLUSTERING:**

**Base:** Python Data Manipulation with Pandas and SQL

**Top:** Diverse AI Applications, Analysis, and Performance

---

**SUMMARY:**

The AI introduces itself, clarifies its capabilities, and offers assistance to the user.

**CLUSTERING:**

**Base:** AI Assistant Initial Interaction and Prompting

**Top:** Diverse AI Applications, Analysis, and Performance

---

**SUMMARY:**

The assistant provides a detailed introduction of Tilley Chemical Co., Inc., including its history, services, and sustainability efforts.

**CLUSTERING:**

**Base:** Chemical Company Profile: Manufacturing, Performance, and Optimization

**Top:** Broad AI Applications, Ethics, and General Knowledge

---

Following CLIO's approach (Tamkin et al., 2024), we implement a hierarchical organization of summaries to improve navigation and comprehension of large conversation datasets. Our process creates a two-level hierarchy with high-level topic clusters containing related lower-level summaries:

1. **Low-level Summary Generation:** We first generate approximately 4,000 low-level summaries using our DP pipeline.
2. **Embedding and Clustering:** These summaries are converted to embeddings and clustered using k-means with $k \approx 70$ to identify broader thematic groups.
3. **High-level Naming:** For each of the 70 high-level clusters, we prompt Gemini to suggest descriptive names based on the contained summaries.
4. **Deduplication and Refinement:** We use Gemini to deduplicate and refine these suggested names, eliminating overlaps and ensuring distinctiveness.
5. **Low-level Assignment:** We represent each high-level name as a center in the embedding space and assign each low-level cluster to its nearest high-level name.
6. **Final Renaming:** Based on the final assignment of low-level clusters, we rename each high-level cluster to better represent its contents.

This hierarchical organization allows users to navigate from broad topics to specific conversation summaries, making the system more useful for exploring large conversation datasets

while maintaining DP guarantees. Some examples of the summary facet, base, and top cluster are demonstrated in Table 4.

