# OpenReview forum: "URANIA: Differentially Private Insights into AI Use"
_colmweb.org/COLM/2025/Conference — COLM 2025_

### Official Review · Reviewer_hkoW · 2025-05-11

**Rating:** 6
**Confidence:** 4
**Ethics Flag:** 1

**Summary:**

This paper presents URANIA, a framework for extracting insights from LLM chatbot interactions with formal differential privacy (DP) guarantees. It uses DP clustering, keyword extraction (frequency, TF-IDF, LLM-guided), and other DP tools for end-to-end privacy. URANIA is evaluated against a non-private baseline, to assess utility and privacy robustness. The paper addresses an important problem with a sound DP-based framework. The end-to-end DP application to this specific problem, including DP keyword set generation, is novel.

**Reasons To Accept:**

* The paper employs formal DP for a deep, insightful, and impactful solution, and offers offers rigorous privacy, unlike heuristic methods.

* It addresses the critical and growing need for privacy-preserving analytics in AI.

* It also provides a reasonable empirical study of the privacy-utility trade-off, despite known challenges in this area.

**Reasons To Reject:**

* The multi-stage pipeline and keyword dependency could be challenging to optimize in practice.

* The proposed approach offers per-conversation privacy, while user-level DP is also important. I'd appreciate more upfront discussion on this and the underlying assumption (correlation between conversations, etc)

* Human evaluation, even small scale, on the summarization results could supplement the LLM-based evaluation.

* The writing could be improved a bit, e.g., I'd try to combine Table 1 and Table 2 results, and link the metrics with the descriptions in line 264-274 better.

---

> ### Author Response · Authors · 2025-05-30
>
> Thank you for your review and recognition of our contribution. We address your feedback below:
>
> **Keyword Generation Challenges:** We agree that improving keyword generation methods represents a key point for enhancing utility. The multi-stage pipeline complexity you mention is also inherent to achieving formal end-to-end privacy guarantees, but we acknowledge this creates optimization challenges that warrant further research. A particularly promising direction for future work would be developing more practical approaches that eliminate the dependency on keyword extraction altogether.
>
> However, we note that our work prioritizes establishing rigorous privacy protections over pipeline efficiency optimization. While computational efficiency is important for practical deployment, our primary contribution is demonstrating that meaningful insights can be extracted from LLM conversations while maintaining formal differential privacy guarantees. Future work could focus on optimizing the computational aspects of our framework, potentially through more efficient DP algorithms or pipeline restructuring, once the fundamental privacy-utility tradeoffs are well-understood.
>
> **User-Level vs. Conversation-Level Privacy:** This is an important distinction. Our current conversation-level approach assumes independence between conversations, which may not reflect real usage patterns. A straightforward extension to user-level DP would involve limiting the number of conversations per user and applying group privacy. While our current high-privacy regimes show significant utility degradation, making the additional privacy cost of group privacy challenging, we believe this limitation becomes more manageable at scale. When working with larger datasets, we can incorporate user-level methods (such as limiting conversations per user) with a proportionally smaller effect on utility. This represents a meaningful direction for future work that requires careful consideration of the privacy-utility tradeoffs.
>
> **Human Evaluation:** We agree that human evaluation would strengthen our assessment. We will try to conduct small-scale human studies comparing purely LLM-based summary quality in future work.
>
> **Writing Improvements:** We will combine Tables 1 and 2 as suggested and better integrate the metric descriptions with the results presentation.  Thank you for the suggestion.

---

> > ### Comment · Reviewer_hkoW · 2025-06-05
> >
> > Thanks for the comment! I'd appreciate it if these discussions could be integrated into the final version especially around the conversation-level privacy and human evaluation.

---

> ### Comment · Area_Chair_4nmX · 2025-06-05
>
> Hello Reviewer! The authors have posted a response to your review. Please respond, at least to let us know *whether or not your views or score have changed*. This will really help the ACs and PCs when we try to assess the paper and your review.

---

### Official Review · Reviewer_qwBB · 2025-05-12

**Rating:** 7
**Confidence:** 4
**Ethics Flag:** 1

**Summary:**

This paper presents URANIA, a framework for generating insights about LLM chatbot conversations with formal differential privacy (DP) guarantees. This work is closely related to Clio, a recent system from Anthropic aiming to generate insights into in-the-wild LLM use while preserving user privacy. Clio doesn't offer formal privacy guarantees and hence this work attempts to bridge this gap. URANIA uses a pipeline that chains together multiple DP algorithms, starting with DP clustering, followed by DP keyword extraction, and finally synthesize summaries of the identified keywords. The evaluation compares URANIA with a simplified version of a Clio-inspired pipeline. Privacy-utility trade-offs were observed, especially in topic coverage.

I overall enjoyed reading this paper and consider it to make a novel and timely contribution. The quality of the work is solid. My main reservation is regarding whether a formal guarantee is really a useful goal to pursue for this problem, as I haven't seen a compelling threat that requires it to be addressed. That said, I think it still deserves further research, and this paper offers a viable solution and valuable insights. Hence, I recommend acceptance.

**Reasons To Accept:**

- It's exploring an intriguing problem of AI for privacy, namely designing an LLM-powered method to gain insights about AI use.
- Offering formal privacy gurantee is a significant improvement compared to the prior work (Clio)
- The paper is well-written. The framework is clearly explained and holds the promises of formal privacy guarantees. The utility evaluation show promising results.

**Reasons To Reject:**

- In the discussion, the drastically reduced topic coverage from 0.723 at low privacy to 0.078 at high privacy was interpreted as "the rare or niche topics are progressively marginalized." However, there is no evidence provided to support that the reduced coverage is related to rare or niche topics rather than missing major topics. Further evidence and justification are needed, or the conclusion should be adjusted.
- This paper seems to focus solely on the privacy leakage risks caused by releasing the clustering results, which is a fine goal to pursue for a paper. However, at a high level, I'm not convinced that a formal privacy guarantee is addressing the most important privacy risks (e.g., a centralized dataset is still required, and it uses a Gemini model to process data, which involves raw data leakage). The empirical privacy evaluation in Appendix E feels somewhat contrived and doesn't demonstrate a significant privacy issue that can be addressed with the private pipeline versus the "public" one.

---

> ### Author Response · Authors · 2025-05-30
>
> Thank you for your thoughtful review and recommendation for acceptance. We address your concerns below:
>
> **Topic Coverage Analysis:** You correctly point out the need for stronger evidence regarding our interpretation of reduced topic coverage. In Table 1, while the number of private summaries decreases from 3,700 to 3,300 (first two rows), the topic coverage drops more dramatically from 0.723 to 0.461. But this statistic alone can not conclude that rare topics are influenced more. We will revise this discussion to be more precise about what our data actually shows.
>
> For example; we would like to replace Line 315 - 317 by the following discussion:
> “Examining the transition from very low privacy (εc = 10.0) to low privacy (εc = 8.0) in Table 1, while the number of private summaries decreases modestly from 3,700 to 3,300 (an 11% reduction), topic coverage drops more dramatically from 0.723 to 0.461 (a 36% reduction). This disproportionate decline suggests that the DP clustering algorithm may systematically exclude certain conversation types even at this early privacy transition, though our current evaluation cannot determine which specific topics are affected. This sharp threshold effect raises important research questions about how different topic types respond to privacy constraints and whether alternative DP approaches might exhibit more gradual degradation."
>
> **Formal Privacy Guarantees vs. Practical Risks:** We appreciate this broader perspective on privacy risks. You're correct that formal guarantees don't address all privacy concerns (e.g., centralized data collection, third-party model processing). However, we believe formal DP guarantees provide valuable complementary protection for the specific risk of information leakage through released summaries. While our empirical evaluation in Appendix E may seem limited, it conceptually demonstrates that even simple attacks may extract meaningful information from non-private pipelines. We view formal guarantees as one layer in a comprehensive privacy strategy, not a complete solution. We will add the discussion on other privacy concerns  in our Discussion and Future Work Section.

---

> > ### Comment · Reviewer_qwBB · 2025-06-09
> >
> > Thanks for the response. The clarifications make sense and I look forward to seeing them in the revised paper. I remain positive about this paper.

---

> ### Comment · Area_Chair_4nmX · 2025-06-05
>
> Hello Reviewer! The authors have posted a response to your review. Please respond, at least to let us know *whether or not your views or score have changed*. This will really help the ACs and PCs when we try to assess the paper and your review.

---

### Official Review · Reviewer_yuSm · 2025-05-18

**Rating:** 7
**Confidence:** 4
**Ethics Flag:** 1

**Summary:**

This paper introduces URANIA, a novel framework for generating insights about LLM chatbot interactions while providing formal differential privacy (DP) guarantees. The authors present a well-structured approach that builds upon previous work (CLIO by Tamkin et al., 2024) but addresses a critical limitation by incorporating rigorous privacy protections.

The quality of the technical work is good. The authors formally define the text summarization problem within a DP framework, develop appropriate algorithms that leverage existing DP tools (clustering, partition selection, and histogram release), and provide mathematical proofs for their privacy guarantees. The experimental evaluation is comprehensive, comparing their approach against a simplified version of CLIO across multiple metrics.

The paper is generally well-written and clear. The problem definition, technical approach, and experimental results are presented in a logical flow. The figures, especially Figure 1, effectively illustrate the proposed method's components. However, some sections could benefit from additional examples to clarify complex concepts.

In terms of originality, URANIA represents an advancement over existing approaches to LLM usage analysis. While the individual DP components used are not novel, their combination and application to text summarization in this context is innovative. The different keyword extraction methods (KwSet-TFIDF, KwSet-LLM, etc.) show thoughtful consideration of various privacy-utility tradeoffs.

As LLMs become increasingly prevalent in daily life, understanding how they are used while protecting user privacy is crucial. The empirical privacy evaluation demonstrating URANIA's robustness compared to non-private alternatives provides compelling evidence of its practical value.

**Questions To Authors:**

1. It would be helpful to include concrete examples of the generated summaries from both URANIA and SIMPLE-CLIO for the same input conversations. This would give readers a better intuition about the quality difference.

2. The authors might consider discussing how their approach could be adapted to handle multimodal conversations that include images, audio, or other media types.

3. The empirical privacy evaluation in Section E is interesting but somewhat limited. Expanding this with more sophisticated attacks would strengthen the privacy analysis.

4. Table 2 mentioned in the text appears to be missing from the main paper (it's referred to on page 7 but appears in Appendix C). Important tables should be included in the main text.

5. The discussion about the degradation in topic coverage at higher privacy levels (lines 309-330) is insightful. The authors might consider expanding on potential solutions to this problem, perhaps by exploring alternative DP clustering approaches.

**Reasons To Accept:**

1. **Novel Privacy-Preserving Framework**: URANIA introduces the a framework for LLM conversation summarization with formal end-to-end differential privacy guarantees. This addresses a significant gap in existing approaches like CLIO.

2. **Theoretical Foundations**: The paper provides solid theoretical foundations, including formal problem statements, privacy proofs, and a principled evaluation methodology that advances the field.

3. **Practical Utility**: Despite the inherent tradeoffs between privacy and utility, the authors demonstrate that URANIA can maintain reasonable utility across various privacy budget configurations, making it practical for real-world applications.

4. **Comprehensive Evaluation**: The evaluation methodology is thorough, using both automated metrics and LLM-based assessments to provide a multifaceted view of the system's performance. The empirical privacy evaluation effectively demonstrates the benefits of DP protections.

5. **Important Application Domain**: As LLMs become ubiquitous tools used by millions of people, understanding usage patterns while protecting privacy is increasingly important. This work addresses a meaningful problem with significant societal implications.

**Reasons To Reject:**

1. **Simplified CLIO Implementation**: Since the authors didn't have access to the original CLIO implementation, they created a simplified version for comparison. This raises questions about how URANIA would compare against the full CLIO system with all its features. A more direct comparison with the original system would strengthen the paper.

2. **Parameter Sensitivity**: While the paper evaluates URANIA across different privacy parameter settings, a more comprehensive analysis of how specific parameters affect various aspects of performance would be valuable. For instance, how do different cluster size thresholds ($\tau$) impact topic coverage and privacy?

3. **Keyword Set Quality**: The keyword set selection appears crucial to URANIA's performance, yet the paper doesn't provide detailed examples of the generated keyword sets or thoroughly analyze how they differ. More examples and analysis here would help readers understand the tradeoffs between different keyword generation approaches.

4. **Scalability and Computational Efficiency**: The paper doesn't adequately address the computational efficiency of URANIA compared to non-private alternatives. As the dataset size grows, how does the performance scale? This is particularly relevant for the DP-K-Means algorithm, which is known to be more computationally intensive.

5. **User-Level DP**: While the authors acknowledge this as future work, the current record-level DP guarantees may be insufficient in practice since individual users typically contribute multiple conversations. A more detailed discussion of the challenges in extending to user-level DP would strengthen the paper.

---

> ### Author Response · Authors · 2025-05-30
>
> We thank you for your thorough review and constructive feedback. We address your questions below:
>
> **1. Concrete Examples of Generated Summaries:**
> Thank you for this valuable suggestion. We will include concrete examples comparing URANIA and SIMPLE-CLIO outputs in the revised manuscript. To illustrate the current limitations, consider these examples from our evaluation:
>
> - **Text:** "The user is asking how to extract the last character of a file in BASH and convert it to hexadecimal."
>     - **Public summary:** "Scripting and Command-Line Task Automation Requests"
>     - **Private summary:** "Software Development, Version Control, and Educational Resources"
>
> - **Text:** "The assistant explains the key differences between reinforcement learning and unsupervised learning, focusing on feedback and goals."
>    - **Public summary:** "Deep Learning, Training, and Applications Explanation"
>    - **Private summary:** "AI Model Training, Deployment, and Management Considerations"
>
> - **Text:** "The user is asking if the assistant speaks Russian and the assistant confirms and offers help."
>    - **Public summary:** "Russian Language Support and Communication"
>    - **Private summary:** "AI-Powered Applications and Open Source Technology"
>
> As these examples demonstrate, public summaries tend to be more specific and directly relevant, while private summaries are often broader and sometimes miss key contextual details. This limitation stems from the privacy constraints that prevent fine-grained keyword extraction—for instance, in the last example case, language-specific terms like "Russian" may not appear in our keyword sets.
>
> **2. Multimodal Conversations:**
> This is an excellent direction for future work. We envision a two-stage approach: first extracting text summaries from multimodal content (e.g., image captions, audio transcriptions), then applying our existing pipeline. However, we agree that establishing a strong non-private baseline for multimodal cases should precede developing the DP version. We will add this in our concluding remarks.
>
> **3. Empirical Privacy Evaluation:**
> We acknowledge that our current attack is relatively simple. As noted in Section 6, attacks against clustering-based algorithms remain underexplored compared to other ML paradigms. Developing more sophisticated attacks to better demonstrate vulnerability is an important area for future research.
>
> **4. Table Placement:**
> Thank you for catching this. We will move Table 2 to the main text in our revision.
>
> **5. Topic Coverage Degradation:**
> You raise an important point about potential solutions. We plan to explore alternative DP clustering approaches and hyperparameter tuning of the DP-k-means implementation to address this limitation.

---

> > ### Comment · Reviewer_yuSm · 2025-06-09
> >
> > Thanks for your responses! Please make sure these are added to the revised manuscript.

---

> ### Comment · Area_Chair_4nmX · 2025-06-05
>
> Hello Reviewer! The authors have posted a response to your review. Please respond, at least to let us know *whether or not your views or score have changed*. This will really help the ACs and PCs when we try to assess the paper and your review.

---

### Decision · Program_Chairs · 2025-07-08

**Decision:**

Accept

**Comment:**

The reviewers are agreed that this paper is ready for publication, addresses an important and timely topic, and makes a useful contribution. I also agree that this paper addresses a very important topic (how to analyze chatbot conversations while preserving privacy) and provides practical methods. The reviewers provide some concrete suggestions for improvement to the camera-ready version of the paper, including additional examples (e.g. keywords) and discussion/framing, that I hope the authors will incorporate.